# Prevalence and Related Factors of Post-ERCP Pancreatitis in Cholangiocarcinoma Patients: A Retrospective Study in Northeast Thailand

**DOI:** 10.3390/jcm14207286

**Published:** 2025-10-15

**Authors:** Tanapoom Karntaumporn, Vasin Thanasukarn, Tharatip Srisuk, Vor Luvira, Theerawee Tipwaratorn, Apiwat Jareanrat, Krit Rattanarak, Khanisara Kraphunpongsakul, Natcha Khuntikeo, Jarin Chindaprasirt, Kulyada Eurboonyanun, Prakasit Sa-Ngiamwibool, Watcharin Loilome, Piya Prajumwongs, Attapol Titapun

**Affiliations:** 1Department of Surgery, Faculty of Medicine, Khon Kaen University, Khon Kaen 40002, Thailand; 2Cholangiocarcinoma Research Institute, Khon Kaen University, Khon Kaen 40002, Thailand; 3Medical Oncology Unit, Department of Medicine, Faculty of Medicine, Khon Kaen University, Khon Kaen 40002, Thailand; 4Departments of Radiology, Faculty of Medicine, Khon Kaen University, Khon Kaen 40002, Thailand; 5Department of Pathology, Faculty of Medicine, Khon Kaen University, Khon Kaen 40002, Thailand; 6Systems Biosciences and Computational Medicine, Faculty of Medicine, Khon Kaen University, Khon Kaen 40002, Thailand

**Keywords:** cholangiocarcinoma, endoscopic retrograde cholangiopancreatography, post-ERCP pancreatitis, preoperative biliary drainage, complications, prevalence, risk factors

## Abstract

**Background/Objectives:** Post-endoscopic retrograde cholangiopancreatography pancreatitis (PEP) is a frequent but insufficiently studied complication in cholangiocarcinoma (CCA). This study aimed to evaluate incidence and risk factors of PEP in CCA. **Methods**: We retrospectively reviewed 148 CCA patients who underwent ERCP between 2019 and 2022. Demographic, clinical, and procedural data were compared between patients with and without PEP, and logistic regression was used to identify independent predictors. **Results**: PEP occurred in 26.4% of patients, mostly aged ≥ 66 years, male, and with perihilar CCA. In PEP cases, metallic stents were frequently used, procedures often exceeded 60 min, and 28.2% developed post-ERCP cholangitis. Hospital stay was ≥4 days in nearly 90% of cases. PEP severity was mild (10.3%), moderate (61.5%), and severe (28.2%). Multivariate analysis showed older age, metallic stent placement, and post-ERCP cholangitis as independent risk factors, while bilirubin ≥ 15 mg/dL was protective. **Conclusions**: PEP occurred in over one-fourth of CCA patients, predominantly of moderate severity. Independent risk factors included older age, metallic stent placement, and post-ERCP cholangitis, whereas bilirubin ≥ 15 mg/dL was protective. Awareness of these factors may aid risk stratification and prevention in this high-risk group.

## 1. Introduction

Cholangiocarcinoma (CCA) is an aggressive epithelial malignancy of the biliary tract with increasing global incidence and one of the highest mortality rates among hepatobiliary cancers [1]. The burden of CCA is particularly pronounced in Southeast Asia, with the highest incidence rates reported in the northeastern region of Thailand [2]. The clinical course of CCA is often insidious and asymptomatic in early stages, leading to delayed diagnosis when the disease is already locally advanced or metastatic [3]. Consequently, only 10–40% of patients are eligible for surgical resection—the sole potentially curative treatment—at the time of diagnosis [4]. For the majority of patients, palliative care remains the mainstay of treatment, focusing on symptom relief and improvement of quality of life. A major clinical challenge in advanced CCA is malignant biliary obstruction, which can cause jaundice, cholangitis, and progressive hepatic dysfunction [5]. Effective biliary drainage is essential not only for symptom control but also for optimizing hepatic function prior to systemic therapy or, in selected cases, delayed surgery [6].

Endoscopic Retrograde Cholangiopancreatography (ERCP) is a crucial procedure for both diagnosing and managing biliary and pancreatic diseases, especially in patients with obstructive jaundice—a common manifestation of CCA [7]. ERCP provides direct ductal visualization, facilitates targeted tissue sampling, and enables biliary decompression through stent placement [8]. In resectable cases, preoperative biliary drainage via ERCP can improve liver function and reduce perioperative risk [9], whereas in unresectable cases, palliative ERCP-guided drainage remains the standard of care [10]. Although ERCP provides significant therapeutic benefit, it is associated with a range of complications, the most common and potentially life-threatening of which is post-ERCP pancreatitis (PEP). The incidence of PEP has been reported to range from 1.6% to 15.7% [11,12,13], with a mortality rate of approximately 0.7% [14,15]. While choledocholithiasis remains the most common indication for ERCP, accounting for about 76% of cases, CCA represents only around 9%. In Northeast Thailand, CCA is a major cause of obstructive jaundice, so ERCP is frequently performed in these patients. The risk of post-ERCP complications, especially PEP, is higher in CCA patients because of tumor-related factors, biliary obstruction, and the complexity of the procedure. PEP has been extensively studied in benign biliary diseases, such as choledocholithiasis and benign biliary strictures [13]. Nevertheless, a significant gap remains in understanding the incidence, risk factors, and clinical consequences of PEP in CCA patients. Most of the existing literature includes only a small proportion of CCA patients, limiting the ability to draw specific conclusions for this population [11,12,13]. Furthermore, few studies have examined the specific clinical context of CCA, including impact of tumor location, stent type, and biliary drainage strategy influence the development of PEP. Data on the consequences of PEP in CCA patients—such as hospital stay, procedural failure, and mortality—remain limited and require further investigation.

Given the expanding role of ERCP in the management of CCA and the potential for serious post-procedural complications, further clarification of the incidence and risk factors of PEP in this specific population is warranted.

This study aimed to determine the incidence of PEP and identify risk factors associated with its occurrence in patients with CCA undergoing ERCP.

## 2. Materials and Methods

### 2.1. Study Design and Patient Population

This retrospective cohort study was conducted to assess the prevalence and risk factors of post-endoscopic retrograde cholangiopancreatography (ERCP) pancreatitis (PEP) in patients with CCA. Data were collected from medical records at Srinagarind Hospital, Khon Kaen University, between August 2019 and March 2022. Among 982 patients who underwent ERCP, 355 were diagnosed with CCA. Inclusion criteria were patients aged over 18 years, with a native major duodenal papilla, and who provided informed consent prior to the procedure. Exclusion criteria included incomplete medical records, previous ERCP, or contraindications to ERCP, such as benign or malignant pancreatic diseases (e.g., acute or chronic pancreatitis, pancreatic neoplasms or cysts), prior gastrectomy, post-ERCP complications other than pancreatitis (e.g., bleeding, cholangitis, or perforation), acute obstructive suppurative cholangitis (AOSC), or pregnancy. After applying these criteria, 148 patients with confirmed CCA who underwent ERCP for diagnostic or therapeutic purposes were included in the final analysis (Figure 1). Ethical approval for this study was obtained from the Khon Kaen University Ethics Committee for Human Research (HE 651319), and the study was conducted in accordance with the principles of the Declaration of Helsinki.

### 2.2. Data Collection

Data were extracted from patient medical records, including demographic information such as age, gender, and body mass index (BMI), as well as underlying medical conditions and the patient’s functional status. In addition, disease-specific data such as the type and location of cholangiocarcinoma, pre-operative total bilirubin levels, and any history of prior pancreatitis were collected. Procedure-related data, including the indication for ERCP (whether diagnostic or therapeutic), the duration of the procedure, and other relevant procedural details, were also recorded. Finally, patient outcomes, including length of hospital stay, treatment costs, and any complications, were documented.

### 2.3. Assessment of Post-ERCP Pancreatitis (PEP)

Post-ERCP pancreatitis (PEP) was defined and classified according to the widely accepted consensus criteria proposed by Cotton et al. [16]. PEP was diagnosed when all three of the following criteria were met:Abdominal pain: New-onset or worsening abdominal pain suggestive of pancreatitis.Laboratory: Serum amylase level elevated to at least three times the upper limit of normal, measured more than 24 h after ERCP.Hospitalization: Hospitalization for more than one night attributable to the pancreatitis episode.

The severity of PEP was graded based on the duration of hospital stay and the need for clinical interventions, as follows:Mild PEP: Hospitalization for 2–3 days without the need for additional interventions.Moderate PEP: Hospitalization lasting 4–10 days.Severe PEP: Hospitalization exceeding 10 days, or the presence of local complications (e.g., pancreatic necrosis, pseudocyst, or other severe complications requiring percutaneous drainage or surgery), consistent with the Cotton criteria [16].

### 2.4. Diagnosis of Other Complications

Cholangitis was diagnosed if a patient exhibited a fever of ≥38 °C, right upper quadrant abdominal pain, and elevated liver enzymes or bilirubin levels [17]. Bile duct perforation was identified either during the endoscopic examination or if the patient developed abdominal pain after the procedure, with radiographic evidence of extraluminal air on X-ray or computed tomography (CT) scan [18].

### 2.5. Outcomes

The primary outcome was the incidence of post-ERCP pancreatitis (PEP) in patients diagnosed with CCA, along with the identification of associated risk factors.

Secondary outcomes included the severity of PEP, as well as the identification of additional risk factors contributing to the development of PEP in this population. Additionally, the study assessed the mortality rate associated with PEP and the incidence of other complications related to post-ERCP pancreatitis in cholangiocarcinoma patients.

### 2.6. Statistical Analysis

Statistical analyses were performed using IBM SPSS Statistics (version 21.0; IBM Corp., Armonk, NY, USA). A *p*-value of <0.05 was considered statistically significant. Descriptive statistics were used to summarize patient characteristics and clinical data. Categorical variables were expressed as frequencies and percentages, and comparisons between groups were made using the Chi-square test or the Fisher-Freeman-Halton exact test, as appropriate. Continuous variables were presented as means ± standard deviations (SD) and compared using the independent samples *t*-test. Variables with a significant association in univariate analysis were further evaluated using multivariate logistic regression to identify independent risk factors.

## 3. Results

### 3.1. Patient Characteristics

The baseline characteristics of the two groups are summarized in Table 1. A total of 148 eligible patients with CCA were included in the study. Among them, 39 (26.4%) developed PEP, whereas 109 (73.6%) did not. The mean age of the cohort was 66 years. Patients aged ≥ 66 years were significantly more likely to develop PEP compared to those aged < 66 years (64.9% vs. 30.6%, *p* = 0.021). No significant difference in gender distribution was observed between the PEP and non-PEP groups (male: 59.0% vs. 65.1%, *p* = 0.493).

Regarding body mass index (BMI), most patients had normal BMI (18.5–24.9 kg/m^2^; 63.5%), followed by overweight/obese (23.0%) and underweight (13.5%). There was no significant association between BMI categories and the development of PEP (*p* = 0.508). Similarly, the presence of underlying diseases such as type 2 diabetes mellitus, hypertension, or both showed no significant difference between the groups (*p* = 0.547).

Most patients had an ECOG performance status of 1 (46.6%) or 2 (40.5%), with no significant difference between groups (*p* = 0.118). However, patients who developed PEP had a significantly higher proportion of pre-ERCP total bilirubin levels < 15 mg/dL compared to the non-PEP group (64.1% vs. 44.0%, *p* = 0.031).

In terms of CCA location, perihilar CCA (pCCA) was the most common (68.3%), followed by distal (15.5%) and intrahepatic CCA (16.2%), with no significant difference between the groups (*p* = 0.985). The rate of sphincterotomy did not differ significantly between groups (*p* = 0.733). Pancreatic duct stent placement was rare in both groups, with no significant difference observed (PEP: 2.6% vs. non-PEP: 3.7%, *p* = 1.000).

Patients who developed PEP were more likely to have metallic stent placement (74.3% vs. 45.0%, *p* = 0.020) and demonstrated a significantly higher rate of technical success (94.9% vs. 77.1%, *p* = 0.013). Procedure duration did not significantly differ between groups (*p* = 0.849).

Importantly, patients in the PEP group had significantly longer hospital stays, with 89.7% hospitalized for ≥4 days compared to 45.9% in the non-PEP group (*p* < 0.001). Additionally, the incidence of post-ERCP cholangitis was significantly higher among patients who developed PEP (28.2% vs. 13.8%, *p* = 0.042). One in-hospital mortality (0.7%) occurred in the PEP group (Table 1).

### 3.2. Severity and Clinical Outcomes of PEP

We analyzed the clinical characteristics of post-ERCP pancreatitis (PEP) patients stratified by severity into mild/moderate and severe groups, as summarized in Table 2. Among the 39 patients with PEP, 11 cases (28.2%) were classified as severe. Most patients were aged ≥ 66 years, particularly in the severe group (81.8%) compared to the mild/moderate group (75.0%). The proportion of male patients was 59.0% overall, slightly higher in the mild/moderate group (64.3%) than in the severe group (45.5%).

Overweight or obese status was observed in 25.6% of the overall cohort, with a higher prevalence in the severe group (45.5%) than in the mild/moderate group (17.9%). Elevated total bilirubin levels (≥15 mg/dL) were also more frequent among patients with severe PEP (54.5%) compared to those with mild/moderate disease (28.6%). Metallic stent placement was recorded in 74.3% of all PEP cases, with a higher rate in the severe group (90.9%) than in the mild/moderate group (67.9%).

Sphincterotomy was performed in 25.6% of cases, with similar distributions between severity groups. ECOG performance status of 3–4 was observed in 15.4% of patients overall, with a higher proportion in the severe group (27.3%) than in the mild/moderate group (10.7%). Most patients (89.7%) had a hospital stay of ≥4 days, including all patients in the severe group. Post-ERCP cholangitis occurred in 28.2% of PEP cases, with comparable rates between the two severity groups.

Although some variables showed numerically higher proportions in the severe group, none of the characteristics were significantly associated with PEP severity based on Chi-square or Fisher’s exact test (all *p* > 0.05).

### 3.3. Risk Factors Associated with PEP

Univariate analysis demonstrated that age ≥ 66 years, total bilirubin ≥ 15 mg/dL, metallic stent placement, technical success, stent type and post-ERCP cholangitis were significantly associated with post-ERCP pancreatitis (PEP) (*p* < 0.05). Other variables, including gender, BMI category, underlying disease (T2DM, hypertension), ECOG performance status, location of CCA, sphincterotomy, stent type (ENBD or plastic stent), procedure time ≥ 1 h, and other baseline clinical factors, did not show statistically significant associations with PEP (*p* > 0.05) (Table 3).

Multivariate logistic regression analysis identified the following as independent risk factors for PEP: age ≥ 66 years (OR 2.89; 95% CI: 1.17–7.17; *p* = 0.022), total bilirubin ≥ 15 mg/dL (OR 0.41; 95% CI: 0.17–0.92; *p* = 0.037), stent type (metallic stent) (OR 3.46; 95% CI: 1.30–7.39; *p* = 0.013), and post-ERCP cholangitis (OR 3.41; 95% CI: 1.21–9.66; *p* = 0.021). Notably, although technical success was significant in univariate analysis, it did not retain statistical significance in the multivariate model (*p* = 0.177) (Table 3).

## 4. Discussion

ERCP plays a pivotal role in managing hepato-pancreato-biliary (HPB) malignancies, particularly CCA, where biliary obstruction is a common and also serious complication. In unresectable cases, ERCP enables effective biliary drainage to relieve jaundice, prevent cholangitis, and preserve liver function for systemic therapy [7]. In potentially resectable disease, preoperative decompression may reduce surgical risks by improving hepatic reserve. ERCP also facilitates tissue sampling and anatomical assessment, contributing to staging and treatment planning. Despite these advantages, ERCP is invasive and carries significant risks, most commonly PEP [19]. Defining these risks in cancer populations, especially CCA, is critical for optimizing procedural safety and outcomes.

In this retrospective study, 148 ERCP procedures were performed in patients with CCA at Srinagarind Hospital, Khon Kaen University, Thailand. The overall incidence of PEP was 26.4%, which is considerably higher than the 1.6–15.7% reported in previous studies [11,12,13,14,15,20]. Most prior research focused largely on benign conditions. For instance, Chi et al. reported a 15.6% incidence of PEP, mainly in patients with choledocholithiasis and cholangitis [20]. Nakeeb et al. studied mixed cohorts of benign and malignant disease, with choledocholithiasis representing 51.4% of ERCP procedures; in that population, mild to moderate PEP occurred in 8% and severe PEP in only 2.2%, with no PEP-related deaths [13]. Similarly, Omar et al. observed a PEP rate of 8.9% in an Egyptian cohort, where choledocholithiasis accounted for 69% of cases and CCA for only 9%. In their study, PEP was mostly mild (63.5%), with 28.8% moderate and 7.7% severe [21]. In contrast, our study focused exclusively on CCA. These patients, particularly those with perihilar tumors (~60% of our cohort), present significant anatomical and technical challenges. Tumor infiltration or compression frequently alters bile duct anatomy and distorts the ampulla of Vater, increasing the difficulty of cannulation and guidewire placement [22]. Management of perihilar CCA often requires advanced techniques such as precut sphincterotomy, multiple guidewires, and bilateral or multisegmental drainage [23]. These complexities likely contributed to the higher incidence and greater severity of PEP observed in our cohort compared with predominantly benign populations.

Despite the higher incidence of PEP in our CCA cohort (26.4%), its severity was predominantly mild to moderate, consistent with previous reports indicating that approximately 80% of cases fall within these categories [13,21]. In our cohort, severe PEP was relatively uncommon, and the mortality rate associated with PEP was only 0.7%, aligning with earlier reports indicating PEP-related mortality rates of less than 1% [12]. Our study identified several independent risk factors for PEP, including older age (≥66 years), metallic stent placement, and post-ERCP cholangitis, whereas higher total bilirubin levels (≥15 mg/dL) appeared to be protective. These findings highlight that risk factors in CCA patients may differ from those reported in predominantly benign populations, underscoring the need for thorough pre-procedural assessment and careful procedural planning. Tailored preventive strategies, particularly for patients with perihilar involvement or complex biliary anatomy, are essential to reduce both the incidence and severity of PEP and to optimize clinical outcomes in this high-risk population.

In our study, patients with CCA aged 66 years and older had an over 80% higher risk of developing PEP compared to those younger than 66 years. This finding is supported by previous research, including a study by Chi et al., who reported that Chinese patients aged 60 years or older undergoing ERCP had a 52.3% higher risk of developing PEP compared to those younger than 60 years [20]. Conversely, Syrén et al. reported a higher risk of PEP among patients younger than 65 years [24]. However, a study by Ergin et al. which investigated PEP risk factors in 2902 Turkish patients undergoing ERCP, found that the incidence of PEP in patients older than 60 years was similar to that in those younger than 60 years [25]. These discrepancies across studies suggest that in elderly patients with CCA, the combination of advanced age and the inherently complex nature of the disease may contribute to an increased risk of developing PEP.

Our results highlight the significant association between metallic stent placement and an increased risk of PEP in patients with CCA. Patients who underwent ERCP with metallic stent insertion had approximately a threefold higher risk of developing PEP compared to those who received either no stent or a plastic stent. These results are supported by previous studies, including a study by Radetic et al., demonstrating that self-expandable metal stent (SEMS) placement during ERCP was associated with a 2.3-fold increased risk of PEP compared to plastic stent placement or no stenting [26]. This risk factor has been further substantiated by reports indicating that SEMS placement is associated with a high incidence of PEP, with rates reported as high as 27% [27,28,29]. The underlying mechanism may involve mechanical and anatomical factors that are particularly relevant in CCA. The expansive radial force of the SEMS can obstruct pancreatic juice outflow by compressing or overlapping the pancreatic duct orifice, especially when the stent is deployed across or near the major duodenal papilla. This compression can narrow or occlude the ductal opening, impair pancreatic drainage, and induce intraductal hypertension, ultimately triggering premature enzyme activation and acute pancreatitis [28]. The risk is further amplified in patients with perihilar or distal CCA, where tumor-related distortion of the biliary anatomy increases the likelihood of papillary irritation. In perihilar CCA (Klatskin tumors), complex strictures often necessitate longer stents across the papilla, while in distal CCA, tumor proximity to the ampulla of Vater requires stent placement directly at or above the papilla, further predisposing to pancreatic duct compression [30]. The large delivery system and high radial force of SEMS exacerbate papillary edema and outflow obstruction—a well-recognized mechanism of PEP [31]. Additionally, patients with extrahepatic CCA typically have non-dilated pancreatic ducts, making them more susceptible to these effects [32]. Post-ERCP cholangitis also appears to contribute to this risk, likely due to severe biliary obstruction, inflammation, and technical complexity during ERCP [33,34].

Moreover, our study identified post-ERCP cholangitis as significantly associated with the development of PEP. CCA patients are often present with high-grade malignant biliary obstruction, subclinical inflammation and distorted biliary anatomy. These conditions may increase the technical complexity of ERCP, thereby predisposing patients to post-procedural complications, including increasing the risk of PEP [33,34]. 

In our study, higher total bilirubin levels (≥15 mg/dL) appeared to act as a protective factor against the development of PEP. This finding is consistent with previous reports, which have suggested that elevated total bilirubin levels at admission are associated with a reduced risk of PEP [35,36]. Boicean et al. suggested that patients without jaundice often have a smaller common bile duct (CBD), which increases the risk of PEP. In contrast, patients with hyperbilirubinemia usually have a dilated biliary system, allowing easier cannulation and reducing pancreatic duct irritation, thereby lowering the risk of PEP.

This study has several strengths. It specifically investigates the prevalence, severity, and procedural risk factors of PEP in patients with CCA, focusing on a large cohort in which perihilar CCA accounted for nearly 60% of cases. Our study also highlights key independent risk factors for PEP, including older age (≥66 years), metallic stent placement, and post-ERCP cholangitis, while identifying higher total bilirubin levels (≥15 mg/dL) as a protective factor. These clinically relevant findings offer a deeper understanding of PEP risk stratification specific to CCA patients. These evidences enhance the reliability of our results which can be applied to similar high-risk patient populations undergoing ERCP.

There are several important limitations to this study. First, it was a single-center retrospective cohort study, which may be subject to selection and information bias. Second, some relevant factors that could influence the risk of PEP—such as detailed cannulation techniques, the use of prophylactic nonsteroidal anti-inflammatory drugs (NSAIDs), and pancreatic stent placement—were not fully evaluated due to limitations in available medical records. To address these gaps, further studies should adopt prospective, multicenter study designs that allow for more comprehensive and standardized data collection. Such approaches would enhance the generalizability of the findings and further clarify the risk factors for PEP in patients with CCA.

## 5. Conclusions

In summary, PEP occurred in over one-fourth of CCA patients. Older age, self-expandable metallic stent placement, and post-ERCP cholangitis were identified as independent risk factors for PEP, whereas elevated total bilirubin levels (≥15 mg/dL) appeared to be protective. These findings emphasize the importance of individualized pre-procedural risk assessment and the selection of appropriate drainage strategies to minimize PEP risk in CCA patients. From a clinical standpoint, careful patient selection, optimal stent choice, and vigilant post-procedure monitoring should be prioritized for high-risk individuals. Future prospective and multicenter studies are warranted to validate these risk factors and to establish standardized preventive protocols tailored to the unique characteristics of CCA-related biliary obstruction.

## Figures and Tables

**Figure 1 jcm-14-07286-f001:**
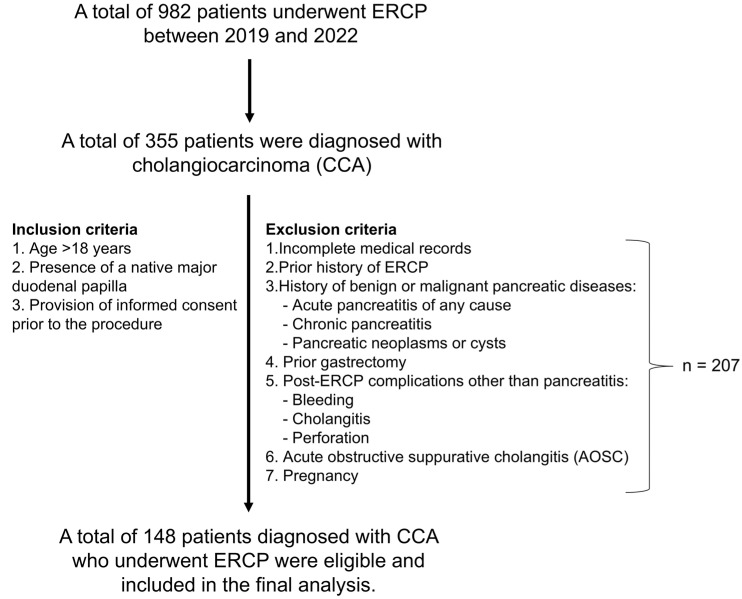
CONSORT diagram illustrating the inclusion and exclusion process of eligible cholangiocarcinoma (CCA) patients for final analysis.

**Table 1 jcm-14-07286-t001:** Comparison of baseline data between the two groups.

Variable	Total (*n* = 148)	Non-PEP *n* = 109 (%)	PEP *n* = 39 (%)	*p-Value*
Age, yr (median (range))	66 (34–89)	64 (34–89)	68 (39–86)	0.021 *
<66	69 (46.6)	57 (52.3)	12 (30.6)	
≥66	79 (53.4)	52 (47.7)	27 (69.4)	
Gender				0.493
Male	94 (63.5)	71 (65.1)	23 (59)	
Female	54 (36.5)	38 (34.9)	16 (41)	
BMI (kg/m^2^), mean ± SD				0.508
<18.5	20 (13.5)	13 (11.9)	7 (17.9)	
18.5–24.9	94 (63.5)	72 (66.1)	22 (56.4)	
>25	34 (23.0)	24 (22)	10 (25.6)	
Underlying disease ^a^				0.547 ^b^
NO	89 (60)	63 (57.8)	26 (66.7)	
T2DM	41 (28)	30 (27.5)	11 (28.1)	
Hypertension	15 (10)	13 (11.9)	2 (5.1)	
Combine	3 (2)	3 (2.8)	0 (0)	
ECOG				0.118
1	69 (46.6)	47 (43.1)	22 (56.4)	
2	60 (40.5)	49 (45.0)	11 (28.2)	
3/4	19 (12.9)	13 (11.9)	6 (15.4)	
Total bilirubin (mg/dL), median (range)	15 (0.2–40.7)	19 (0.2–40.4)	14.9 (0.3–40.7)	0.031 *
<15	73 (49.3)	48 (44)	25 (64.1)	
≥15	75 (50.7)	61 (56)	14 (35.9)	
Location of CCA				0.985
iCCA	24 (16.2)	18 (16.5)	6 (15.4)	
pCCA	101 (68.2)	74 (67.9)	27 (69.6)	
dCCA	23 (15.6)	17 (15.6)	6 (15.4)	
Sphincterotomy				0.733
No	113 (76.4)	84 (77.1)	29 (74.4)	
Yes	35 (23.6)	25 (22.9)	10 (25.6)	
Pancreatic duct stent				1.000 ^b^
No	143 (96.6)	105 (96.3)	38 (97.4)	
Yes	5 (3.4)	4 (3.7)	1 (2.6)	
Stent type				0.020 ^b,^*
No stent	29 (19.6)	25 (22.9)	4 (10.3)	
ENBD	23 (15.5)	19 (17.4)	4 (10.3)	
Plastic stent	18 (12.2)	16 (14.7)	2 (5.1)	
Metallic stent	78 (52.7)	49 (45)	29 (74.3)	
Technically success				0.013 *
No	27 (18.2)	25 (22.9)	2 (5.1)	
Yes	121 (81.8)	84 (77.1)	37 (94.9)	
Total procedure time (min.), median (range)	60 (18–140)	60 (18–120)	60 (20–140)	0.849
<60	55 (37.2)	41 (37.6)	14 (35.9)	
≥60	93 (62.8)	68 (62.4)	25 (64.1)	
Length of stay (days), median (range)	4 (2–36)	3 (2–36)	7 (3–21)	<0.001 *
<4	63 (42.6)	59 (54.1)	4 (10.3)	
≥4	85 (57.4)	50 (45.9)	35 (89.7)	
Post-ERCP complication (cholangitis)				0.042 *
No	122 (82.4)	94 (86.2)	28 (71.8)	
Yes	26 (17.6)	15 (13.8)	11 (28.2)	
Mortality (in hospital)	148	0	1 (0.7%)	-

^a^ In some cases, the information regarding underlying diseases was not available. ^b^ The expected counts in some cells being less than 5, the Fisher–Freeman–Halton Exact Test was used instead of the Chi-Square Test to assess the association between the variables in this table. * *p* < 0.05 was considered statistically significant.

**Table 2 jcm-14-07286-t002:** Characteristics of PEP patients stratified by severity.

Variable	Total PEP *n* = 39 (100%)	Mild/Moderate *n* = 28 (71.8%)	Severe *n* = 11 (28.2%)
Age ≥ 66 years, *n* (%)	30 (76.9)	21 (75)	9 (81.8)
Male sex, *n* (%)	23 (59.0)	18 (64.3)	5 (45.5)
Overweight/Obese, *n* (%)	10 (25.6)	5 (17.9)	5 (45.5)
pCCA, *n* (%)	27 (69)	20 (71)	7 (63)
Total bilirubin ≥ 15 mg/dL, *n* (%)	14 (35.9)	8 (28.6)	6 (54.5)
Metallic stent, *n* (%)	29 (74.3)	19 (67.9)	10 (90.9)
Sphincterotomy, *n* (%)	10 (25.6)	7 (25.0)	3 (27.3)
ECOG 3–4, *n* (%)	6 (15.4)	3 (10.7)	3 (27.7)
Length of Stay ≥ 4 days, *n* (%)	35 (89.7)	24 (85.7)	11 (100)
Post-ERCP cholangitis, *n* (%)	11 (28.2)	8 (28.6)	3 (27.3)

**Table 3 jcm-14-07286-t003:** Risk factors of post-ERCP pancreatitis.

Variable	Univariate Analysis			Multivariate Analysis		
	OR	95% CI	*p*-Value	OR	95% CI	*p*-Value
Age, yr (Median)						
<66	1			1		
≥66	3.65	1.59–8.42	0.002 *	2.89	1.17–7.17	0.022 *
Gender						
Male	1			-		
Female	1.30	0.61–2.75	0.493	-	-	-
BMI (kg/m^2^)						
<18.5)	1			-		
18.5–24.9	0.57	0.20–1.59	0.284	-	-	-
>25	0.77	0.24–2.51	0.670	-	-	-
Underlying disease ^a^						
No	1			-		
T2DM	0.888	0.39–2.03	0.780	-	-	-
Hypertension	0.373	0.08–1.77	0.214	-	-	-
Combine ^b^	-	-	-			
ECOG score						
1	1			-		
2	0.48	0.21–1.10	0.082	-	-	-
3–4	0.99	0.33–2.94	0.980	-	-	-
Total bilirubin (mg/dL), median (range)						
<15	1			1		
≥15	0.44	0.21–0.94	0.034 *	0.41	0.17–0.92	0.037 *
Location of CCA						
iCCA	1			-		
pCCA	1.09	0.39–3.05	0.863	-	-	-
dCCA	1.06	0.29–3.93	0.932	-	-	-
Sphincterotomy						
No	1			-		
Yes	1.16	0.49–2.70	0.733	-	-	-
Stent type						
Non-metallic stent ^c^	1			1		
Metallic stent	3.55	1.58–7.99	0.002 *	3.46	1.30–7.39	0.013 *
Technically success						
No	1			1		
Yes	5.51	1.24–24.46	0.025 *	12.51	0.59–18.07	0.177
Total procedure time (min), median (range)						
<60	1			-		
≥60	1.08	0.50–2.30	0.849	-	-	-
Post-ERCP cholangitis						
No	1			1		
Yes	2.46	1.01–5.97	0.046 *	3.41	1.21–9.66	0.021 *

^a^ In some cases, the information regarding underlying diseases was not available. ^b^ fewer cases. ^c^ The non-metallic stent group includes patients with no stent, ENBD, or a plastic stent., as shown in Table 1. ** p* < 0.05 was considered statistically significant.

## Data Availability

The datasets created and analyzed in this study are not publicly accessible because of ethical agreements concerning participant privacy. However, data sharing options can be discussed by contacting the corresponding author.

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
