# Peer review of "Prevalence and Related Factors of Post-ERCP Pancreatitis in Cholangiocarcinoma Patients: A Retrospective Study in Northeast Thailand"

_jcm, 2025, doi:10.3390/jcm14207286_

Round 1

Reviewer 1 Report

Comments and Suggestions for Authors

The article Prevalence and Related Factors of Post-ERCP Pancreatitis in Cholangiocarcinoma Patients provides particularly valuable information on a niche topic. Specifically, there are very few studies in the literature presenting data on post-ERCP pancreatitis in patients with cholangiocarcinoma. Recommendations:

  1. The first chapter is well written and presents clear and useful background information for the reader.
  2. Present more clearly the criteria for defining post-ERCP pancreatitis (not just by citation).
  3. The flow chart in the Results section should be moved to Materials and Methods.
  4. The results are excellent! The number of patients is sufficient and the statistical analyses are of good quality.
  5. The Discussion section correctly and comprehensively explains the results obtained, with concrete justifications from the literature.
  6. The Conclusions can be expanded.

Author Response

Reviewer 1

Comments and Suggestions for Authors

The article Prevalence and Related Factors of Post-ERCP Pancreatitis in Cholangiocarcinoma Patients provides particularly valuable information on a niche topic. Specifically, there are very few studies in the literature presenting data on post-ERCP pancreatitis in patients with cholangiocarcinoma. Recommendations:

  1. The first chapter is well written and presents clear and useful background information for the reader.

Author response: We sincerely appreciate the reviewer’s positive feedback and are pleased that the Introduction was found to be clear and informative.

  1. Present more clearly the criteria for defining post-ERCP pancreatitis (not just by citation).

Author response: We thank the reviewer for this valuable suggestion. In response, we have revised the Methods section (2.3 Assessment of Post-ERCP Pancreatitis) – page number 4, and line 123-137 to present the diagnostic criteria more clearly. Specifically, we now:

  1. List the three diagnostic criteria explicitly in a numbered format.
  2. Emphasize that all three criteria must be met for the diagnosis of PEP.
  3. Separate the severity grading into bullet points for clarity.

These changes ensure that the criteria are fully described within the text, rather than relying solely on the citation, improving readability and transparency.

  1. The flow chart in the Results section should be moved to Materials and Methods.

Author response: We agree with the reviewer’s comment. The flow chart originally presented in the Results section has been moved to the Materials and Methods section, under “2.1 Study Design and Patient Population.”- page number 3, and line 97-100. The content and paragraph order in this section have also been slightly revised to improve the logical flow and readability.

  1. The results are excellent! The number of patients is sufficient and the statistical analyses are of good quality.

Author response: We sincerely thank the reviewer for the positive comments and appreciation of our results. We are pleased that the number of patients and the quality of the statistical analyses were found to be appropriate.

  1. The Discussion section correctly and comprehensively explains the results obtained, with concrete justifications from the literature.

Author response:  We thank the reviewer for the positive comments and for recognizing the clarity and comprehensiveness of our Discussion section.

  1. The Conclusions can be expanded.

Author response:  We thank the reviewer for the valuable suggestion. In response, we have expanded the Conclusions section to include additional clinical implications and future perspectives in page number 11, and line 363-372

5. Conclusions

In summary, PEP occurred in over one-fourth of CCA patients. Older age, self-expandable metallic stent placement, and post-ERCP cholangitis were identified as independent risk factors for PEP, whereas elevated total bilirubin levels (≥15 mg/dL) appeared to be protective. These findings emphasize the importance of individualized pre-procedural risk assessment and the selection of appropriate drainage strategies to minimize PEP risk in CCA patients. From a clinical standpoint, careful patient selection, optimal stent choice, and vigilant post-procedure monitoring should be prioritized for high-risk individuals. Future prospective and multicenter studies are warranted to validate these risk factors and to establish standardized preventive protocols tailored to the unique characteristics of CCA-related biliary obstruction.”.

Reviewer 2 Report

Comments and Suggestions for Authors

Dear Authors,  Your retrospective cohort study on post-ERCP pancreatitis in cholangiocarcinoma patients provides valuable insights into an understudied area, with sound methodology and relevant clinical implications. This reviewer recommends minor revisions to address inconsistencies, typos, and language polishing.  Please make the following changes and resubmit the revised manuscript along with a point-by-point response.  

Required revisions:

  1. The Methods section (line 90) states "A total of 135 patients," but the Abstract, Results (line 152), and Figure 1 indicate 148 patients. Please correct this to ensure consistency (likely update Methods to 148).  
  2. The Methods section (line 87) lists the period as August 2019 to March 2023, but the Abstract and Results (line 149) state 2019-2022. Align all references to the correct timeframe.  
  3. The sentence "PEP occurred in 26.4% of patients, mostly aged ≥66 years, male, and with perihilar CCA" is duplicated (lines 23–24). Remove or rephrase one instance for conciseness.   

Typos and spelling errors:

o Section 3.2 heading (line 194): Change "everity" to "Severity."

o Ensure consistency in abbreviations and capitalization (e.g., "Post-ERCP cholangitis" vs. "post-ERCP cholangitis").

o Table 3: Verify confidence intervals in the multivariate analysis (e.g., for total bilirubin ≥15 mg/dL, ensure CI is 0.17–0.92, not misformatted).   

Minor clarifications:

o Methods (line 122): Clarify if "local complications (e.g., pseudocyst or necrosis)" includes all relevant severe PEP markers per Cotton criteria.

o Discussion: Expand briefly on why metallic stents increase PEP risk in CCA specifically (e.g., reference tumor location more explicitly if data supports).

o References: Ensure all DOIs are active and formatting is uniform (e.g., some are lowercase). No major issues noted.

Comments on the Quality of English Language

The English is generally clear but requires editing for grammar, awkward phrasing, and run-on sentences to improve flow and professionalism.  

Examples:

o Introduction (lines 61–64): Split long sentences for readability (e.g., combine or rephrase the discussion of PEP incidence and indications).

o Discussion (line 299): Change "suggested" to "suggest" for tense consistency. o Discussion (lines 278–283): Rephrase repetitive ideas about risk factors and tailored strategies for conciseness (e.g., "These findings suggest that while PEP is more frequent in CCA patients...").

o General: Avoid overusing passive voice; ensure non-idiomatic phrases are revised (e.g., line 329: "One possible explanation... is that a smaller common bile duct (CBD), often observed in patients without jaundice, is linked to a higher risk of PEP" could be smoothed). Recommend review by a native English speaker or professional editing service.

Author Response

Comments and Suggestions for Authors

Dear Authors,  Your retrospective cohort study on post-ERCP pancreatitis in cholangiocarcinoma patients provides valuable insights into an understudied area, with sound methodology and relevant clinical implications. This reviewer recommends minor revisions to address inconsistencies, typos, and language polishing.  Please make the following changes and resubmit the revised manuscript along with a point-by-point response.  

Required revisions:

  1. The Methods section (line 90) states "A total of 135 patients," but the Abstract, Results (line 152), and Figure 1 indicate 148 patients. Please correct this to ensure consistency (likely update Methods to 148).  

Response: We sincerely thank the reviewer for this helpful comment. The number of patients has been corrected to 148 throughout the manuscript to ensure consistency. In addition, Figure 1 has been relocated to the Methods section (“2.1. Study Design and Patient Population,” page 3, lines 100–102) as suggested of reviewer 1.

  1. The Methods section (line 87) lists the period as August 2019 to March 2023, but the Abstract and Results (line 149) state 2019-2022. Align all references to the correct timeframe.  

Response: We appreciate the reviewer’s valuable comment. The study period has been revised to “August 2019 to March 2022” throughout the Abstract, and Methods sections to maintain consistency.

  1. The sentence "PEP occurred in 26.4% of patients, mostly aged ≥66 years, male, and with perihilar CCA" is duplicated (lines 23–24). Remove or rephrase one instance for conciseness.   

Response: We thank the reviewer for this valuable comment. The duplicated sentence (“PEP occurred in 26.4% of patients, mostly aged ≥66 years, male, and with perihilar CCA”) has been removed in the revised manuscript to improve clarity and conciseness.

Typos and spelling errors:

o Section 3.2 heading (line 194): Change "everity" to "Severity."

Response: We appreciate the reviewer’s helpful comment. The heading in Section 3.2 has been corrected to “Severity” as suggested.

o Ensure consistency in abbreviations and capitalization (e.g., "Post-ERCP cholangitis" vs. "post-ERCP cholangitis").

Response: We thank the reviewer for this helpful comment. The use of capitalization for terms such as “post-ERCP cholangitis” has been carefully checked throughout the manuscript. The capitalization style follows standard English grammar conventions, with uppercase letters used only when the term appears at the beginning of a sentence. Consistency has been ensured accordingly.

o Table 3: Verify confidence intervals in the multivariate analysis (e.g., for total bilirubin ≥15 mg/dL, ensure CI is 0.17–0.92, not misformatted).   

Response: We thank the reviewer for this helpful observation. The confidence intervals for all hazard ratios presented in the multivariate analysis (Table 3) have been thoroughly checked and revised where necessary to ensure both numerical accuracy and proper formatting.

Minor clarifications:

o Methods (line 122): Clarify if "local complications (e.g., pseudocyst or necrosis)" includes all relevant severe PEP markers per Cotton criteria.

Response: Thank you for your comment. We have clarified that “local complications” include all relevant severe PEP markers as defined by the Cotton criteria, including pancreatic necrosis, pseudocyst, and other serious complications requiring percutaneous drainage or surgery. The revised text has been updated accordingly.

o Discussion: Expand briefly on why metallic stents increase PEP risk in CCA specifically (e.g., reference tumor location more explicitly if data supports).

Response: We sincerely thank the reviewer for this valuable suggestion. We have expanded the Discussion to explain in greater detail why metallic stents increase the risk of PEP specifically in CCA patients, with reference to tumor location and relevant anatomical mechanisms. The revised section (page 10, lines 315–332) now reads as follows:

The underlying mechanism may involve mechanical and anatomical factors that are particularly relevant in CCA. The expansive radial force of the SEMS can obstruct pancreatic juice outflow by compressing or overlapping the pancreatic duct orifice, especially when the stent is deployed across or near the major duodenal papilla. This compression can narrow or occlude the ductal opening, impair pancreatic drainage, and induce intraductal hypertension, ultimately triggering premature enzyme activation and acute pancreatitis [28]. The risk is further amplified in patients with perihilar or distal CCA, where tumor-related distortion of the biliary anatomy increases the likelihood of papillary irritation. In perihilar CCA (Klatskin tumors), complex strictures often necessitate longer stents across the papilla, while in distal CCA, tumor proximity to the ampulla of Vater requires stent placement directly at or above the papilla, further predisposing to pancreatic duct compression [30]. The large delivery system and high radial force of SEMS exacerbate papillary edema and outflow obstruction—a well-recognized mechanism of PEP [31]. Additionally, patients with extrahepatic CCA typically have non-dilated pancreatic ducts, making them more susceptible to these effects [32]. Post-ERCP cholangitis also appears to contribute to this risk, likely due to severe biliary obstruction, inflammation, and technical complexity during ERCP [33,34].

o References: Ensure all DOIs are active and formatting is uniform (e.g., some are lowercase). No major issues noted.

Response: We appreciate the reviewer’s valuable comment. All DOIs in the References have been verified to be active, and their formatting has been made uniform throughout the manuscript. For References 14 and 19, no DOI could be identified; however, both works are well-recognized and highly cited publications, and we have retained them in the Reference list due to their reliability and relevance.

Comments on the Quality of English Language

The English is generally clear but requires editing for grammar, awkward phrasing, and run-on sentences to improve flow and professionalism.  

Response: We sincerely thank the reviewer for the helpful suggestion regarding language editing. The manuscript has been carefully reviewed and edited for grammar, phrasing, and flow by Professor              Ross H. Andrews through the Publication Clinic at Khon Kaen University to improve clarity and professionalism.

Examples:

o Introduction (lines 61–64): Split long sentences for readability (e.g., combine or rephrase the discussion of PEP incidence and indications).

Response: We thank the reviewer for this valuable suggestion. We have revised this part of the Introduction to improve readability by splitting the long sentences and clarifying the discussion of PEP incidence and indications. The revised paragraph (page 3, lines 66–75) now reads as follows:

The incidence of PEP has been reported to range from 1.6% to 15.7%[11-13], with a mortality rate of approximately 0.7% [14,15]. While choledocholithiasis remains the most common indication for ERCP, accounting for about 76% of cases, CCA represents only around 9%. In Northeast Thailand, CCA is a major cause of obstructive jaundice, so ERCP is frequently performed in these patients. The risk of post-ERCP complications, especially PEP, is higher in CCA patients because of tumor-related factors, biliary obstruction, and the complexity of the procedure. PEP has been extensively studied in benign biliary diseases, such as choledocholithiasis and benign biliary strictures [13].  Nevertheless, a significant gap remains in understanding the incidence, risk factors, and clinical consequences of PEP in CCA patients

o Discussion (line 299): Change "suggested" to "suggest" for tense consistency.

Response: We thank the reviewer for this helpful comment. We have corrected the verb tense for consistency as suggested. The word “suggested” has been changed to “suggest” in the Discussion section (page 10, line 304).

 o Discussion (lines 278–283): Rephrase repetitive ideas about risk factors and tailored strategies for conciseness (e.g., "These findings suggest that while PEP is more frequent in CCA patients...").

Response: We appreciate the reviewer’s valuable comment. We have revised this part of the Discussion to reduce repetition, enhance conciseness, and improve flow while maintaining the original meaning. The revised text (page 9–10, lines 282–294) now reads as follows:

Despite the higher incidence of PEP in our CCA cohort (26.4%), its severity was predominantly mild to moderate, consistent with previous reports indicating that approximately 80% of cases fall within these categories [13,21]. In our cohort, severe PEP was relatively uncommon, and the mortality rate associated with PEP was only 0.7%, aligning with earlier reports indicating PEP-related mortality rates of less than 1% [12]. Our study identified several independent risk factors for PEP, including older age (≥66 years), metallic stent placement, and post-ERCP cholangitis, whereas higher total bilirubin levels (≥15 mg/dL) appeared to be protective. These findings highlight that risk factors in CCA patients may differ from those reported in predominantly benign populations, underscoring the need for thorough pre-procedural assessment and careful procedural planning. Tailored preventive strategies, particularly for patients with perihilar involvement or complex biliary anatomy, are essential to reduce both the           incidence and severity of PEP and to optimize clinical outcomes in this high-risk population.

o General: Avoid overusing passive voice; ensure non-idiomatic phrases are revised (e.g., line 329: "One possible explanation... is that a smaller common bile duct (CBD), often observed in patients without jaundice, is linked to a higher risk of PEP" could be smoothed). Recommend review by a native English speaker or professional editing service.

Response: We thank the reviewer for this valuable suggestion. We have revised the sentence to improve clarity, reduce the use of passive voice, and ensure smoother phrasing. The revised text (page 10, lines 341–344) now reads as follows:

Boicean et al. suggested that patients without jaundice often have a smaller common bile duct (CBD), which increases the risk of PEP. In contrast, patients with hyperbilirubinemia usually have a dilated biliary system, allowing easier cannulation and reducing pancreatic duct irritation, thereby lowering the risk of PEP.”

Reviewer 3 Report

Comments and Suggestions for Authors

The article " Prevalence and related factors of post-ERCP pancreatitis in cholangiocarcinoma patients" evaluates incidence and risk factors of post-endoscopic retrograde cholangiopancreatography pancreatitis (PEP) in cholangiocarcinoma (CCA) patients. Study finds higher prevalence of PEP in over one-fourth of CCA patients, predominantly of moderate severity. Independent risk factors included older age, metallic stent placement, and post-ERCP cholangitis, whereas bilirubin ≥15 mg/dL was protective. PEP being the most common complication of ERCP which is usually seem in about 10% of patients. So, this is well-known knowledge in clinical practice, and this study validates the knowledge in new cohort of patients.

Interestingly North-eastern Thailand being most epidemic region for CCA around the globe, this particular study cohort representing the area truly reflects the prevalence and factors accepted need to be well-acknowledged.  I do agree with authors that awareness of these factors may aid risk stratification and prevention in this high-risk group. The study has several limitations which authors have duly accepted and explained in brief.

It is a well-studied and well-written manuscript. However, few suggestions for improvement of the manuscript are as follows.

  1. Please add information of patients cohort in the title. As this is not the first study and patient cohort representing North-east Thailand adds so much value being the most epidemic region from the world. The prevalence/factors associated carries much information than other cohorts.
  2. Line 23-24 contains duplicate sentences. Please thoroughly proofread and edit the manuscript to eliminate redundancy and improve clarity and flow.

Author Response

Reviewer 3

Comments and Suggestions for Authors

The article " Prevalence and related factors of post-ERCP pancreatitis in cholangiocarcinoma patients" evaluates incidence and risk factors of post-endoscopic retrograde cholangiopancreatography pancreatitis (PEP) in cholangiocarcinoma (CCA) patients. Study finds higher prevalence of PEP in over one-fourth of CCA patients, predominantly of moderate severity. Independent risk factors included older age, metallic stent placement, and post-ERCP cholangitis, whereas bilirubin ≥15 mg/dL was protective. PEP being the most common complication of ERCP which is usually seem in about 10% of patients. So, this is well-known knowledge in clinical practice, and this study validates the knowledge in new cohort of patients.

Interestingly North-eastern Thailand being most epidemic region for CCA around the globe, this particular study cohort representing the area truly reflects the prevalence and factors accepted need to be well-acknowledged.  I do agree with authors that awareness of these factors may aid risk stratification and prevention in this high-risk group. The study has several limitations which authors have duly accepted and explained in brief.

It is a well-studied and well-written manuscript. However, few suggestions for improvement of the manuscript are as follows.

  1. Please add information of patients cohort in the title. As this is not the first study and patient cohort representing North-east Thailand adds so much value being the most epidemic region from the world. The prevalence/factors associated carries much information than other cohorts.

Response: We agree in this point, and revised the manuscript title from “Prevalence and related factors of post-ERCP pancreatitis in cholangiocarcinoma patients” to “Prevalence and related factors of post-ERCP pancreatitis in cholangiocarcinoma patients: A retrospective study in Northeast Thailand” to include information about the patient cohort as suggested.

  1. Line 23-24 contains duplicate sentences. Please thoroughly proofread and edit the manuscript to eliminate redundancy and improve clarity and flow.

Response:  We have carefully proofread and edited the manuscript to eliminate the duplicate sentences on lines 23–24 of Abstract section– page number 1, and line 27.
